# Durability and Property Study of Decade Old Crumb Rubber Concrete Cored Specimens

**DOI:** 10.3390/ma15165490

**Published:** 2022-08-10

**Authors:** Fuqiang Duan, Han Zhu, Yasser E. Ibrahim, Musa Adamu

**Affiliations:** 1School of Civil Engineering, Tianjin University, Tianjin 300350, China; 2Engineering Management Department, College of Engineering, Prince Sultan University, Riyadh 11586, Saudi Arabia

**Keywords:** crumb rubber concrete (CRC), cored specimens, long-aged, semi-circular disk, accelerated re-bar corrosion test, carbonation test

## Abstract

Crumb rubber concrete (CRC) is a concrete that contains rubber crumbs. This article presents a study of three experiments on long aged CRC specimens that were cored from a decade old CRC bridge deck in Tianjin, China. The three experimental tests conducted were: (1) the flexural stress–strain test on semi-circular disk specimens; (2) the accelerated steel-rebar corrosion test and (3) the carbonation test. In addition, the in situ carbonation test was also carried out on the CRC bridge deck. The flexural stress–strain test results showed that the CRC semi-circular disk specimens exhibited a ductile pattern and high-energy absorbing capacity with its flexural tensile strength being at 5 MPa and the flexural modulus of 10 GPa. The steel corrosion rust rate via the calculation of steel mass loss before and after the test in the accelerated steel-rebar corrosion test remained extremely low. The carbonation test results showed that in comparison with the prediction of two popular carbonation models, the carbonation in the CRC bridge deck took place at a much slower rate during the last 13 years. All of the results obtained in this study are reported for the first-time and indicate that these CRC cored specimens exhibit good mechanical properties and excellent durability characteristics after a decade in service, which may provide the technical knowledge for the possible future application of CRC in concrete constructions.

## 1. Introduction

Crumb rubber concrete (CRC) is a composite material by adding rubber crumbs into Portland cement concrete. Rubber crumbs are made by shredding old automobile tires. In this sense, CRC is a “green” material. The early study on CRC can be traced back to Savas et al. [1] in the early 1990s on its mechanical properties. Topcu [2] studied the effect of the particle size of rubber crumbs on strength through tests. Results showed that coarse tire chips reduced the compressive strength more than fine tire chips. Al-akhras and Smadi [3] found that rubber crumbs improved the compressive and flexural strength of cement mortar, but had a negative impact on flowability and prolonged the initial and final setting time. Zhu et al. [4,5,6] carried out a series of experimental studies on cement paste, mortar, and concrete containing rubber crumbs that indicated the benefit of high energy absorption in comparison with ordinary Portland cement concrete with enhanced bending toughness and reduced the elastic modulus. Xue et al. [7] found that adding both the expansion agent and rubber crumbs into cement mortar improved cracking resistance. An experiment investigating the vibrating of a T-shaped cantilever beam was conducted by Xu et al. [8], where they found that CRC had a much higher damping capacity than ordinary concrete.

With respect to durability, rubber crumbs can reduce noticeable early drying shrinkage in CRC [9]. Singh et al. [10] reported that adding rubber improved the water permeability of CRC. Gesolgu et al. [11] studied the influence of the rubber particle size on the freezing resistance of pervious concrete. The results they obtained showed that the thermal fatigue performance of the pervious concrete mixed with rubber was significantly improved and it can withstand 300 freeze–thaw cycles.

It is fair to say that after 40 years of study, the material properties of CRC have been both well-researched and understood. The publications on CRC may exceed thousands if not ten thousands worldwide. The overall description of CRC is that, in comparison with ordinary concrete, it has high deformation/strains, energy absorption, ductility, damping capacity, cracking resistance, and excellent freeze–thaw resistance, and its benefit can go on to a long list. Despite all the good things about CRC, the real world application of CRC is very limited because CRC suffers from two weaknesses. One is that adding rubber crumbs into concrete will reduce the compressive and flexural strength that is vital to the concrete industry. The other is that rubber tends to float upward in a fresh mix when slump is high (say 10 cm or more), which causes the segregation phenomenon in the CRC mix. For the strength reduction, this can be compensated for by adding more cement and reducing the water–cement ratio at a cost hike so that the strength design requirement can be met. These days, CRC can readily be made at a compressive strength of 60 MPa or more. Regarding the rubber float-upward problem, the remedy is to add additives that increase the viscous level of the fresh concrete mix. One method is to add Superabsorbent polymer (SAP), a water-absorbing polymer, into CRC, which was successfully employed in a bridge deck construction in Henan, China in 2018 with the CRC slump being 18 cm and higher and concrete pumping facility being used in pouring CRC to the bridge deck [12]. With the two remedies above-mentioned, the CRC engineering applications seem ready to move forward. However, concrete material engineers are worried that another critical issue that is aging effect on CRC performance, especially since, currently, no such study on long-aged CRC has been found in the public literature, and little knowledge is known in this regard.

In early August 2006, a CRC bridge deck was constructed in a suburb of Tianjin, China, as shown in Figure 1. In June 2017, the bridge was inspected and six specimens of 150 mm in diameter with a thickness varying from 70 mm to130 mm were cored from the deck, as shown in Figure 2.

Six core specimens from the aged CRC bridge deck were further investigated with respect to: (1) the spilt strength test; (2) the CRC density measure by the wax-immersion method; (3) the rubber content measure and calculation; (4) crack observation; and (5) the layered distribution of rubber crumbs with the computerized image scanning method. The details of the investigation can be found in [13] by the current authors. It was found that the CRC specimens were in good condition and retained excellent material integrity, with the flexural strength converted from the split strength data exceeding its original magnitude. This means that aging actually facilitates the increase in the CRC material strength.

This article continues the study in [13] by conducting the three further experiments and two carbonation models. The first experiment was to measure the flexural stress-strain responses by the so-called semi-circular disk bending test. The aim was to see whether aging will change the characteristics of CRC, or not, as aforementioned. The second was to first externally implant a steel rebar into CRC cored specimens and then put them into salt water by inserting an electric voltage field so that the salt water would permeate through the concrete to reach the rebar for steel corrosion to occur. Then, the rust level and steel mass loss were measured to see how well the CRC could perform in resisting chlorite ion penetration. The third experiment was the carbonation test for the purpose of seeing how well the CRC withstood the carbonation of a decade long period. In addition, two carbonation models were employed to simulate the CRC carbonation behavior. Additionally, a carbonation onsite test was also carried out on the CRC bridge deck.

## 2. The Significance

So far, most of the experiments on CRC specimens have been conducted in a laboratory. The experimental tests conducted in this article were carried out on specimens cored from a decade old in-service CRC bridge deck for the purpose of studying long-term CRC behaviors with respect to the durability and mechanical properties. Such a study has not been found in any of the public literature that has explored how well CRC withstands a decade of tear and wear with respect to its long-age effect on the mechanical properties and durability as well as to provide the technical knowledge for possible future application of CRC in concrete constructions.

## 3. Experimental Tests

The primary information on materials such as CRC mix proportion, cored specimens, etc. can be found in [13]. The CRC bridge was constructed in 2006, the specimens cored from the deck in 2017, and the experimental tests in this article were carried out in from 2019 to 2020.

### 3.1. Semi-Circular Disk Bending Test

The semi-circular disk bending (SCDB) experimental test is frequently used to measure the flexural tensile stress–strain response and the flexural strength of the asphalt concrete specimens. The experimental process is to apply a vertical load on the top center of the semi-circular disk specimen until fracture failure takes place and the specimen is broken into two pieces. The maximum load and deformation are recorded to obtain the stress–strain response and tensile strength. The fracture energy can also be obtained by calculating the area of the load deformation curve. The SCDB experiment originated in the category of rock mechanics in which Wei et al. [14] studied sedimentary rocks to measure the fracture toughness. However, most SCDB studies can be found in asphalt concrete research. Al-Qudsi et al. [15] obtained semi-circular asphalt concrete load–displacement curves under low temperature. Kran et al. [16] and Ven et al. [17] studied the fatigue cracking of asphalt mixture by the SCDB test method.

It is well-known that the tensile property of Portland cement concrete is obtained by the beam type of bending test. A limited number of reports have explored SCDB on Portland cement concrete. Razmi and Mirsayar [18] showed that in SCDB evaluation, the fiber reinforced concrete specimens had a better crack resistance than ordinary concrete. In this study, the cored specimens were all of a cylindrical shape, leaving SCDB as the only option to evaluate long-aged CRC concrete flexural stress-stain properties and failure strength. The SCDB specimens were made by first splitting a round specimen into two halves of semi-circular disks (see Figure 3a), and then polishing the split face of the semi-circular disk and attaching strain gauges, as shown in Figure 3b. Figure 4 shows a diagram of the SCDB test setup.

### 3.2. Accelerated Steel Rebar Corrosion Test

The corrosion of steel rebar in steel reinforced concrete takes place when chlorite ions react with the steel rebar. In a natural environment, the chlorite ions permeate very slowly inside concrete, and the corrosion or rust rate of rebar in concrete is generally low, taking months, years, or even decades to be observed. As such, accelerated steel rebar corrosion tests in a laboratory environment have been developed to gear up the steel-rusting speed [19,20,21,22] as a way to evaluate the concrete ability in resisting chlorite ion permeation. The method employed in this study was the electric accelerated corrosion test as described: putting rebar concrete specimens in salt water in a semi-immersion manner with the steel rebar being above the salt water, which also serves as an anode of an external electric voltage loop. The semi-immersed rebar concrete specimen also sits on steel that serves as the cathode (under salt water). Then, a constant voltage is applied so that an electric field is produced and exerted on the rebar concrete. The chlorite ions in the salt water begin penetrate the concrete at a much quicker speed as they are repelled by the electric field (see the sketch of the test set-up in Figure 5). When chlorite ions react with steel, rust takes place. After a certain time, this test is finished and the rebar inside the concrete specimens will be taken out, polished to remove rust, and the weight of the steel rebar will be measured. The relative loss of the steel weight before and after the test is seen as an indicator of how the specimen concrete performs in resisting chlorite corrosion. The lower the loss in the steel weight, the better corrosion resistance in the concrete.

The specimens in this test are usually a steel reinforced beam. However, since only cored specimens were available in the study, the steel rebar had to be implanted by drilling a hole into a semi-circular disk CRC specimen, inserting the steel bar into the hole and using cement to “glue” the steel to the disk. In this way, six specimens were made (see Figure 6a,b), and then the tests were carried out accordingly.

### 3.3. Carbonation Test

Concrete is a strong alkali material with a pH greater than 12.5, but carbon dioxide in air will slowly permeate from the surface to the interior of the concrete and react with calcium hydroxide, which results in the decrease in pH and weakens the alkaline protection of steel rebar in concrete, which is called the concrete carbonation phenomenon. Therefore, the carbonation test, which measures the carbonation depth in concrete, is considered as an important durability assessment. Houst and Wittmann [23] studied the influence of the porosity and concrete water content on the diffusion of CO_2_ in hardened cement slurry. Saetta et al. [24] established the differential relationship between water diffusion, CO_2_ airflow diffusion, and temperature change in the concrete carbonation process. In addition to the experimental work, many scholars have conducted theoretical analyses to model or simulate the carbonation process in concrete for the purpose of estimating the time of carbonation [25,26,27,28] as a way of predicting the life of the reinforced concrete service time.

The carbonation test is a relatively simple experiment that sprays a phenolphthalein indicator solution into a pre-made concrete facade along the depth direction, producing a purple-red color zone when it meets un-carbonized concrete and a colorless zone when it meets carbonized concrete. The thickness of the colorless zone is measured as the carbonation degree (see Figure 7). The details can be found in the China Standard for test methods of long-term performance and durability of ordinary concrete (GB/T 50082-2009).

## 4. Experimental Results

### 4.1. Semi-Circular Disk Bending Test

#### 4.1.1. Test Program

This test was conducted according to the German Standard for fiber reinforced Concrete (DBV 1998) [29]. The dimensions of the specimens was a diameter of 150 mm and a thickness of 40 mm. Three strain gauges were attached in parallel to the center of the bottom of each specimen and a TDS-530 strain data acquisition instrument was used to record the bending strain. The load was in the speed control mode with the speed rate of 0.03 mm/second by a microcomputer controlled electronic universal testing machine with the total force reactively recorded.

#### 4.1.2. Test Results and Analysis

While the strains were recorded in the test, the corresponding bending stress was also calculated by the following equation:σ = 3 FL/2BR^2^(1)
where σ is the tensile stress (MPa); F is the reactively recorded total load (N); L is the support span (mm), which is 120 mm; B is a semi-circular disk width (mm); R is the semi-circular radius (mm). In total, three specimens were performed and Figure 8 presents the averaged corresponding bending stress–strain curves. It can be seen that the three CRC specimens exhibited a good ductile type of material characteristics with the tensile ultimate strain being about 6.0% and the flexural stress strength being about 5.0 MPa.
(2)$$A=\int_{0}^{\varepsilon }\sigma \left(\varepsilon \right)d\varepsilon$$

For the purpose of estimating the bending strain energy, point B and point C were further introduced and labelled into the three curves in Figure 8 for the purpose of indicating the elastic limit point and the ultimate limit point of the stress–strain responses in Figure 8, respectively, which is sketched in Figure 9. Accordingly, *σ*_B_ and *ε*_B_ represent the elastic limit strain and stress, respectively. *σ*_C_ and *ε*_C_ represent the ultimate limit strain and stress, respectively.

It can be observed in Figure 9 that the curve of section OB was basically linear [30], and the strain increased significantly after passing through point B. This means that the specimens produced a large value of deformation in their non-elastic responses. The areas under the curve, namely, the bending strain energy, can be obtained by integrating the curve, and the strain energy formula is as follows:

The areas under the OB section are defined as the CRC elastic strain energy AE, and the area under the BC section is defined as the CRC on-linear strain energy AP. The total strain energy (AE + AP) was used to illustrate the energy absorption capacity of CRC, and the results are tabulated in Table 1.

#### 4.1.3. Flexural Modulus

The flexural modulus is an important material property in concrete pavement engineering. Following the stress–strain curves in Figure 8, the flexural modulus E_r_ of the CRC specimens can be deduced and expressed in terms of Equation (3):E_r_ = 23 L^3^ × (F_0.5_ − F_0_)/(1296 J × (δ_0.5_ − δ_0_))(3)
where F_0.5_ and F_0_ are the semi-final load and initial load (N); δ_0.5_ and δ_0.5_ correspond to the dial meter reading of F_0.5_ and F_0_, respectively. L is the distance between the pedestals of the specimen; J is the moment of inertia of the specimen section in which J = 1/12 bh^3^. Three corresponding calculated E_r_ are given in Table 2.

### 4.2. Accelerated Steel-Rebar Corrosion Test

#### 4.2.1. Test Program

The photo of the actual test setup given in Figure 10 and Figure 11 shows how the rebar-implanted specimen sat on two steel wires that served as the cathode, as aforementioned. This means that there was a gap between the specimen and the container. A total of six specimens, three semi-circular disk specimens and three quarterly-circular disk specimens, were used in the test with the specimen thickness of 40 mm. Holes of 6–8 mm in diameter were drilled in the specimens at a point of 1~1.5 cm away from the edge, and a 6 mm diameter and 10 cm long steel bar coated with cement was inserted into the drilled hole. The salt water in the tank was a 3.5 wt% NaCl solution with the level being maintained at an approximately 1 cm height. The accelerated steel-rebar corrosion test is not a standardized test, and 3.5% wt% NaCl is what has been used in previous studies. A power supply consists of a voltage output of 12 V, with a maximum output current of 50 A. The test time was 5 days for three specimens and 10 days for one specimen.

#### 4.2.2. Test Results

At the completion of the test, the implanted steel bars were taken out of specimens (see Figure 12), washed and polished by wire brush, and sandpapered to remove the concrete residue and part of the rust on the surface. They were then soaked in 10% dilute sulfuric acid for 15 min, cleaned with tap water, and polished with clean sandpaper until the rebar was close to the background color. Finally, the steel bars were dried in the oven and weighed on an electronic balance. The corrosion rust rate of steels is expressed by the weight loss percentage in comparison with its weight before and after rust, is expressed in Equation (4) and its values are given in Table 3 for Specimens #1–#6. Table 3 also includes the results from Chen [31], labeled as S-1-1 and S-1-3.
The corrosion rust rate = (w_0_ − w)/w_0_ × 100%(4)
where w_0_ and w are the steel rebar weight before and after rust, respectively.

### 4.3. Carbonation Test

#### 4.3.1. Test Program

The indoor carbonation test was conducted according to the GB-T50082-2009 standard [32]. The cored CRC specimen was divided into four pieces of 10 cm length, 2 cm width, and 3 cm thickness, as shown in Figure 13. A solution of 1% phenolphthalein in 99% ethyl alcohol was sprayed on the freshly cracked facade. The average of the colorless depth was taken as the carbonation depth, as given in Table 4.

In addition to the carbonation test on the broken CRC cored specimens, the on-site test on seven locations on the CRC deck was also carried out by drilling holes with a diameter of 15 mm and a depth of 10 mm on the deck, and the phenolphthalein indicator solution was sprayed into the holes (see Figure 14). Then, the depth from the boundary between purple red and colorless to the bridge deck was measured and recorded three times. The average of the three measurements was taken as the carbonation depth are also given in Table 4.

#### 4.3.2. Carbonation Depth Prediction Model

The main objective of the carbonation test is to see what the current level of the carbonation depth is in the concrete from the surface onward to the concrete inside and the possible future development. As such, many scholars have proposed theoretical models to predict the carbonation depth in the future. Among them, the two models developed by Niu [33] and Zhang [34] have received extensive attention because the former prediction is seen as too big (overestimate), and the latter is seen as too small (underestimate). These two models were also employed in this article to serve as the upper and lower limits to provide a relative comparison between the two models and the test data obtained in this study.

The Niu carbonation model of concrete under stress is expressed as follows:X = *K_com_* × *t*^0.5^(5)
*K_com_* = *h_mc_k_c_k_r_k_b_k_j_k_w/c_k_CO2_k_e_k_f_k_s_*(6)
where X is the carbonation depth; *K_com_* is the concrete carbonation comprehensive coefficient, which reflects environmental conditions, concrete material property, and stress conditions, and the unit is mm/*t*^0.5^; t is the natural carbonation time (in year); *h_mc_* is the model adjusting constant and the regression constant without stress; *k_c_* is the cement dosage influence coefficient; *k_r_* is the cement type influence coefficient; *k_b_* is the curing method influence coefficient, which is 1.0 for standard curing and 1.85 for steam curing; *k_j_* is the angular correction coefficient, 1.4 for the angular part, and 1.0 for the non-angular part; *k_w/c_* is the water–cement ratio influence coefficient; *k_CO2_* is the natural carbonation in the atmospheric environment; *k_e_* is the influence coefficient of ambient temperature and humidity; *k_f_* is the concrete quality influence coefficient; *k_s_* is the concrete stress level influence coefficient.

In this study, *t* = 13 (it means the time of 13 years from 2006–2019); *h_mc_* = 1.645; *k_c_* = 253C^−0.964^ in which C is the cement weight per cubic meter concrete. In the study, C = 481 kg and *k_c_* = 0.657; *k_r_* was 1.0 for ordinary cement, and 1.35 for slag cement. In this study, *k_r_* = 1. Additionally, *k_b_* = 1 and *k_j_* = 1.4. In this study, *k_w/c_* = (W/(0.3822 × C))^0.5^ = 1.023, where W/C = 0.4. *k_CO2_* = 1. *k_e_* = 2.564(1 − RH)RH × T^0.25^, and T and RH are the annual mean temperature and annual mean humidity, respectively. In this study, T = 11.6, RH = 0.62, and *k_e_ =* 1.115. *k_f_* = 57.94/f_cuk_ − 0.76 where *f_cuk_* is the standard value of the compressive strength of concrete in MPa, and in this study, *f_cuk_* = 40 and *k_f_* = 0.689; *k_s_* = 0.0149 + 27.729s − 62.224s^2^ + 57.439s^3^ − 18.224s^4^,and s = 0.8 and *k_s_* = 4.319. With all of these parameters assigned a value, the outcome of Equation (5) is: X = 10.934 mm.

The prediction model of concrete carbonation depth X proposed by Zhang is as follows:X = *K_w_*(0.1 × T)^0.713^(RH2 − 1.98RH + 1.896)(C_0_/0.03)^0.5^(15.806/f_cuk_ + 0.215)*t*^0.42^(7)
where *K_w_* is the influence coefficient of the concrete curing condition with the outdoor being = 1.0 and indoor being = 1.87. In this study, *K_w_* = 1.0. The T and RH values were taken as given in Niu’s model: T = 11.6 and RH = 0.62. *C_o_* is the CO_2_ volume concentration in the carbonation environment and *C_o_* = 0.039% in this study. Again, *f_cuk_* = 40 MPa in this study. *t* is the service time of concrete in years and *t* = 13. With all of the parameters assigned a value, Zhang’s model in Equation (7) yielded at X being 2.106 mm (also see Table 4).

## 5. Discussion

Observing Figure 8, it can be seen that the ultimate flexural tensile strain could reach 600 × 10^−6^, which was much higher than that of ordinary concrete. Additionally, the flexural tensile strength exceeded 5 MPa. Moreover, the specimens underwent considerable deformation before failure and showed an obvious inelastic strain development stage. The total area under the curve (AE + AP) was more than eight times that of the AE value. All indicate that the long aged CRC had good material strength, bending toughness, and large deformation properties.

For typical ordinary concrete pavement, its flexural elastic modulus is about 30 GPa, which is considered as “rigid”. Bai [35] conducted a four-point bending experiment on ordinary concrete and the flexural elastic modulus was about 34 GPa. However, with the addition of rubber into the concrete, the flexural elastic modulus dropped to 20 GPa or lower. Typically, concrete will become more rigid as it ages. However, for the values of the flexural elastic modulus given in Table 3, the long-aged CRC still maintained a low value. 

Lowing the flexural elastic modulus may have a profound influence on the airfield pavement design. In the current design cases, the flexural elastic modulus is set at a default value of 36 GPa. The study conducted by Li et al. [36] indicates that because of its low value of flexural elastic modulus in the 20–30 GPa range, CRC could reduce the thickness of the concrete airfield pavement by 9% and increase the fatigue pavement life in a noticeable way. Table 2 in this article shows that the values decreased to the 10–20 GPa range at a decade old time frame, which shows that CRC may be plausible in providing more benefits to concrete airfield pavement design.

Table 3 shows the corrosion rust rate from the current study. Taking the average, the 5 day and 10 day number was 0.84 and 4.11, respectively. For a comparison purpose, Table 4 also includes the results of S-1-1, which is ordinary concrete and S-1-3, which is CRC with 100 kg rubber per cubic meter, close to the 92 kg rubber of CRC specimens in this study. The numbers in Table 4 show that after a decade of aging, the CRC specimens in this study still exhibited excellent corrosion resistance, which was comparable to that obtained in Chen’s study. 

Regarding long-aged concrete carbonation, Chen et al. [37] made a set of ordinary concrete blocks with the dimensions of 1500 × 1200 × 600 and compressive strengths from 37.8 MPa to 76.4 MPa. These blocks were placed outdoor at the city of Zhengzhou in the central part of China. The carbonation depth was measured at a sequential time interval and the last one was conducted in 2015, which was 14 years later. The values of the carbonation depth at 14 years was 10.21 mm for 37.8 MPa and 0.88 mm for 76.4 MPa, respectively. For the current study, as displayed in Table 4, the averaged carbonation depth for the seven on-site measurements was 1.33 mm and 1.16 mm for the four cored specimens. In addition, as displayed in Table 4, the Niu model prediction was 10.21 mm, which was extremely higher that the numbers in the CRC. In most cases, the Zhang model prediction had a low limit, but it also exceeded the measured value in the CRC.

## 6. Conclusions

This article mainly carried out three types of experimental tests including semi-circular disk bending (SCDB) tests, carbonation tests, and electric accelerated corrosion tests on decade-old cored CRC specimens. The results can be concluded for this decade long-aged CRC specimens as follows: (1) Maintaining good material properties such as high energy absorption capacity, good flexural strength, high deformation, and low flexural tensile module; (2) good ion chloride corrosion resistance; and (3) low carbonation depth. This indicates that CRC possesses good long-aged durability that is comparable to good quality ordinary concrete. It should be mentioned that the results obtained here were obtained using a small number of samples, which did not possess any statistical significance.

## Figures and Tables

**Figure 1 materials-15-05490-f001:**
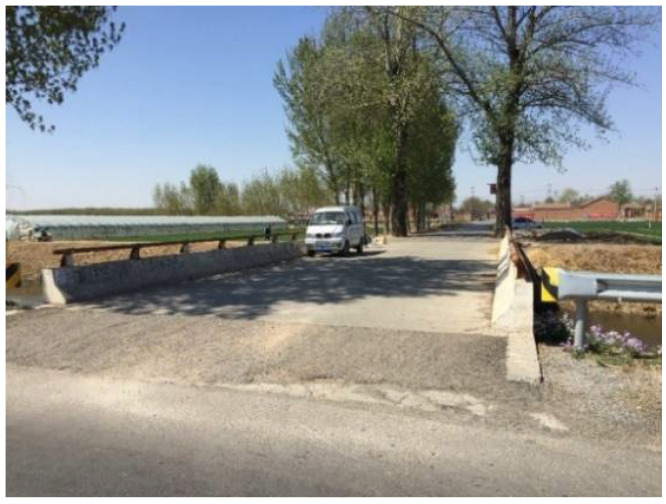
A bridge with a CRC deck built in 2006 in Tianjin, China.

**Figure 2 materials-15-05490-f002:**
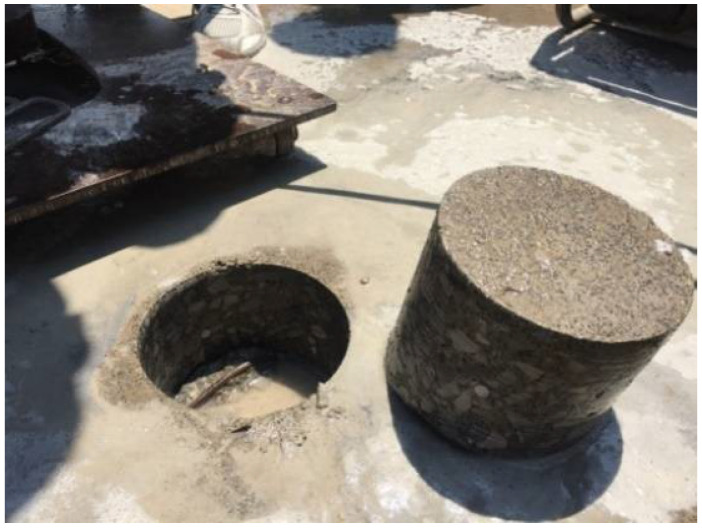
The cored CRC specimens in 2017.

**Figure 3 materials-15-05490-f003:**
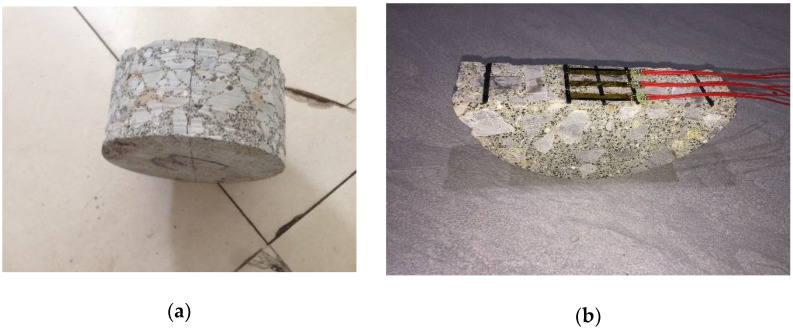
(**a**) Splitting a cored specimen. (**b**) Semi-circular disk with attached strain gauges.

**Figure 4 materials-15-05490-f004:**
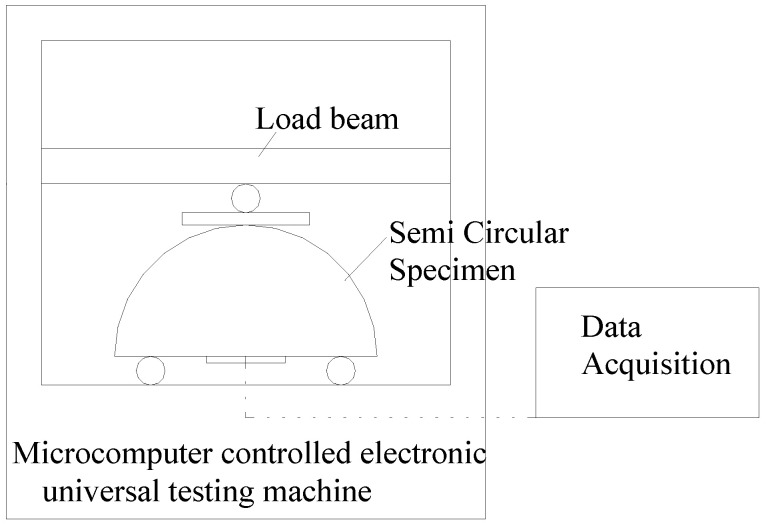
The concept of the SCDB test setup.

**Figure 5 materials-15-05490-f005:**
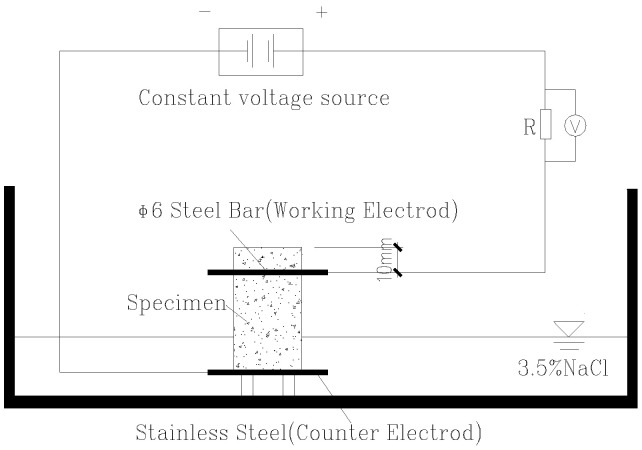
Accelerated steel rebar corrosion test setup.

**Figure 6 materials-15-05490-f006:**
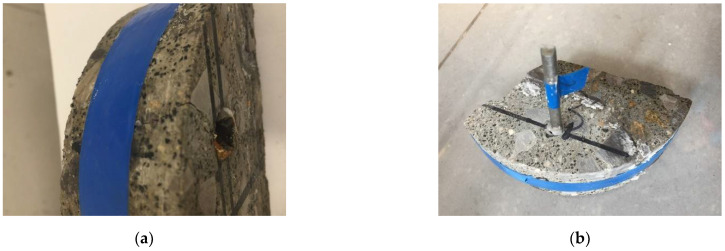
(**a**) Drilling a hole in the cored specimen; (**b**) implanting the steel rebar.

**Figure 7 materials-15-05490-f007:**
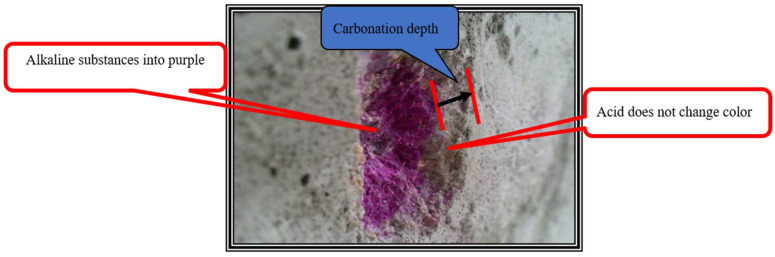
The carbonation test.

**Figure 8 materials-15-05490-f008:**
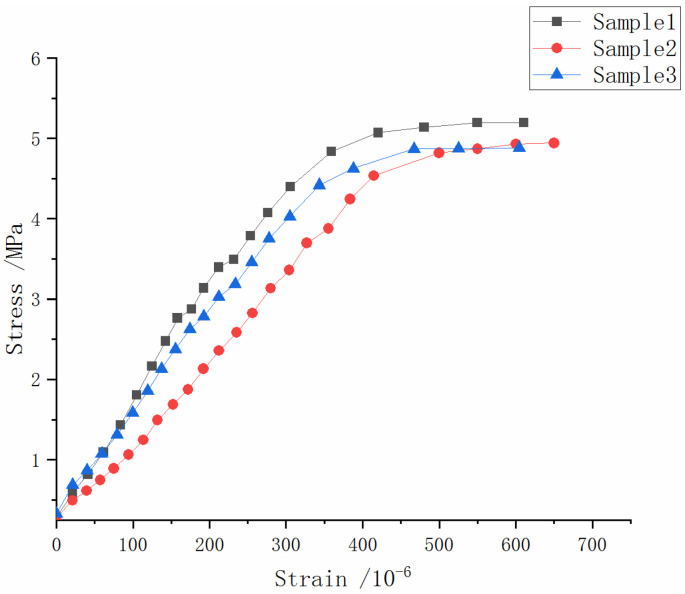
The flexural stress–strain responses of three specimens in the SCDB test.

**Figure 9 materials-15-05490-f009:**
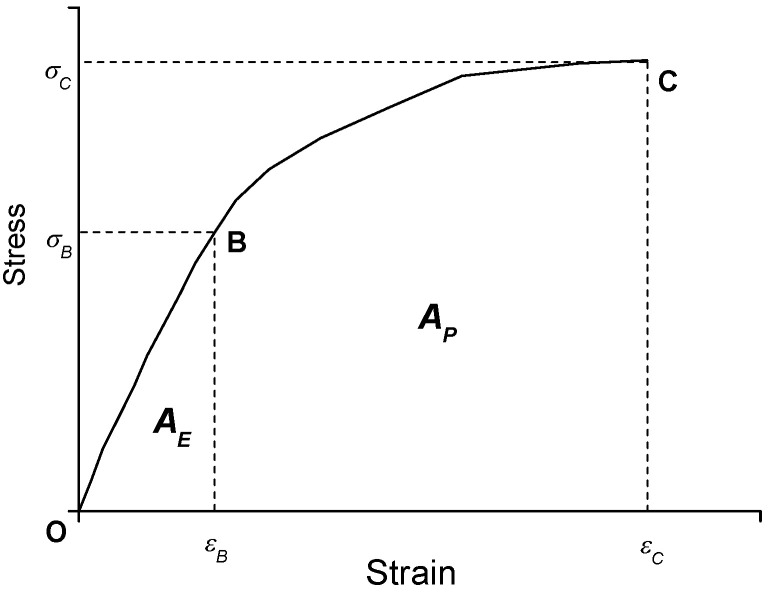
The linear and nonlinear responses of SCDB test.

**Figure 10 materials-15-05490-f010:**
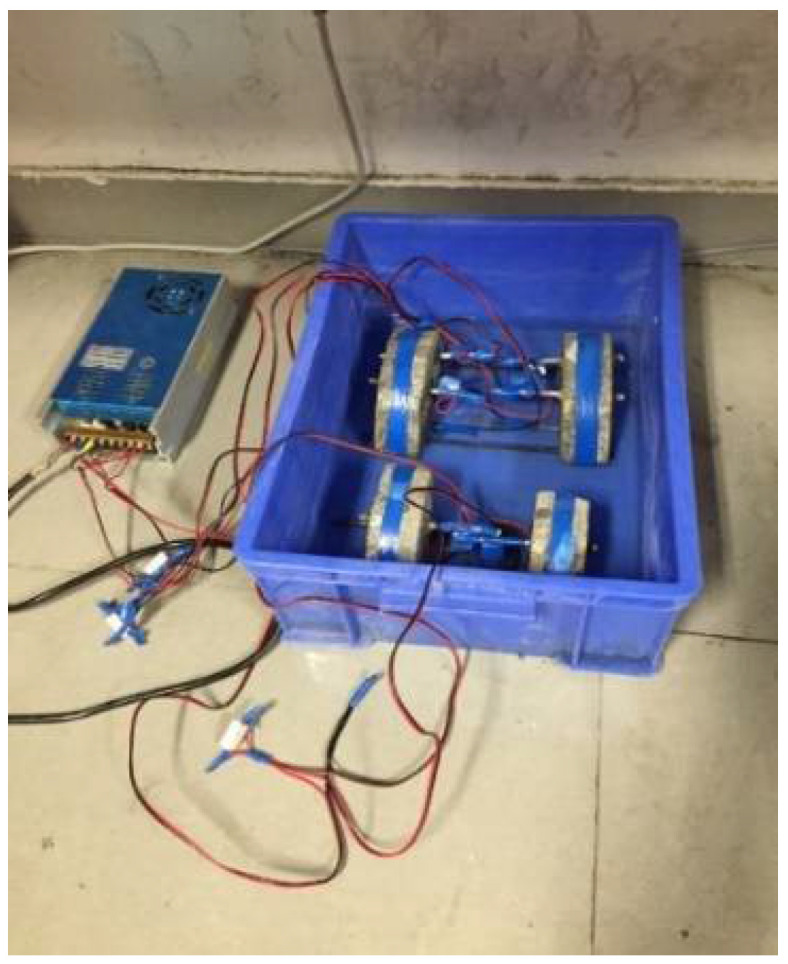
A photo of the setup of the accelerated steel-rebar corrosion test.

**Figure 11 materials-15-05490-f011:**
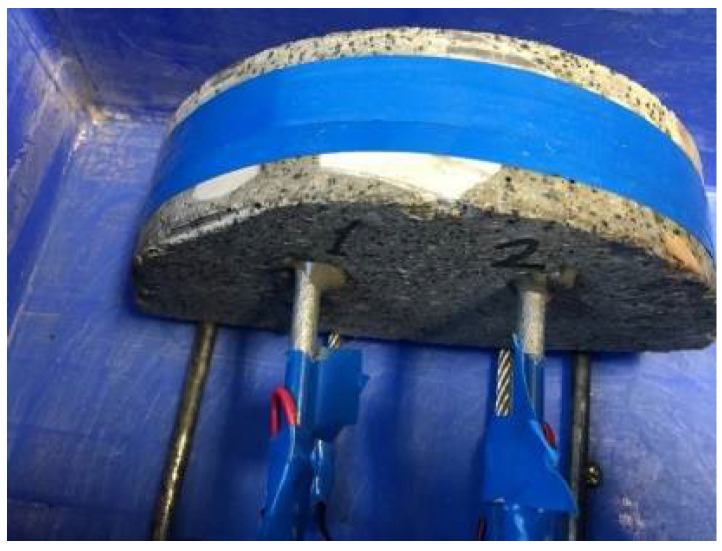
The rebar-implanted specimen sat on two steel wires.

**Figure 12 materials-15-05490-f012:**
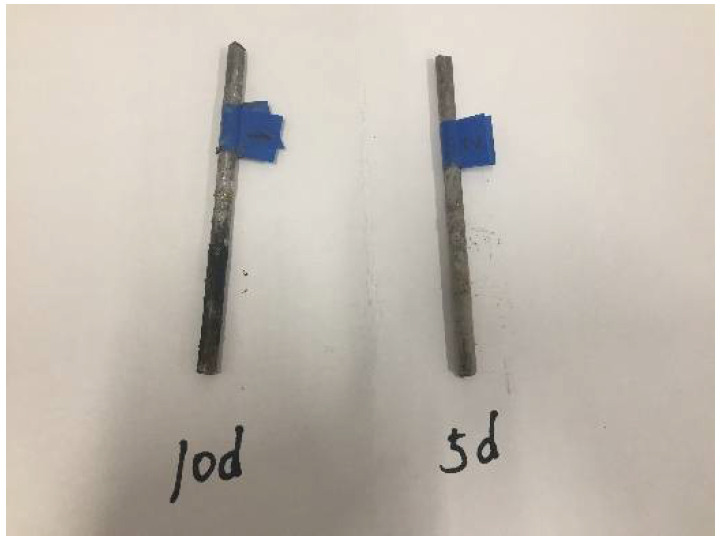
The rusty steel bar at 5 days and 10 days of the accelerated corrosion test.

**Figure 13 materials-15-05490-f013:**
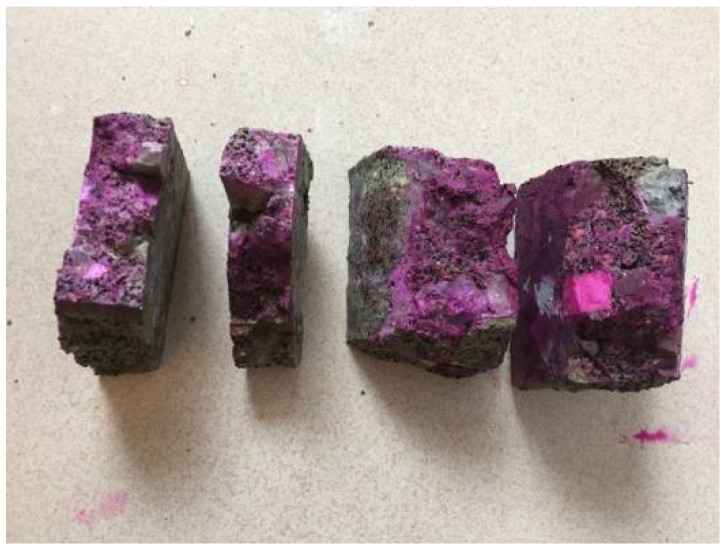
The carbonation test on four CRC cored specimens.

**Figure 14 materials-15-05490-f014:**
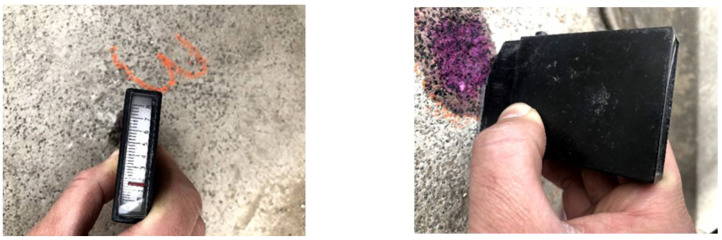
The in situ carbonation tests on the CRC bridge deck.

**Table 1 materials-15-05490-t001:** The elastic and non-elastic strain energy calculation in the SCDB test.

Number	AE/Pa	AP/Pa	(AE + AP)/AE	Er/GPa
Specimen-1	276.8	2004.6	8.24	13.47
Specimen-2	248.9	1846.0	8.42	13.03
Specimen-3	218.4	1653.4	8.57	10.14

**Table 2 materials-15-05490-t002:** The calculated flexural modulus of Er (GPa) for the three specimens.

	Specimen-1	Specimen-2	Specimen-3
E_r_	13.47	13.03	10.14

**Table 3 materials-15-05490-t003:** The corrosion rust rate (%).

Specimen#	5 d	10 d
1	-	5.08
2	-	4.06
3	-	3.20
4	0.66	-
5	1.07	-
6	0.79	-
S-1-1	3.06	6.02
S-1-3	1.69	5.39

**Table 4 materials-15-05490-t004:** The measured carbonation depths with the predictions (mm).

On-Site Test	1.25	1.75	2.00	0.50	1.50	1.35	1.00
Specimen test	1.52	0.84	0.63	1.66			
Niu-model	10.93						
Zhang-model	2.10

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
