# Peer review of "Durability and Property Study of Decade Old Crumb Rubber Concrete Cored Specimens"

_materials, 2022, doi:10.3390/ma15165490_

Round 1

Reviewer 1 Report

This article investigates the durability of decade-old crumb rubber concrete cored specimens. The important discussion sections are not clearly presented and they should improve extensively. This article requires more improvement to enhance its quality. The comments are listed below.

 1. Abstract: discuss one or two lines about the current issues of this research, the solution and what is the need for this research. The description of the investigation is shallow and should be elaborated. I suggest putting numbers within brackets (1), (2) and (3)….“The three experimental tests are: 1 the flexural stress-strain test on semi-circular disk specimens; 2 the accelerated steel-rebar corrosion test and 3 the carbonation test. In addition, the in-situ carbonation test was also carried out on the CRC bridge deck.”

2. Introduction: I suggest adding a few lines about the issues due to crumb rubber and how it affects the environment and current methods to minimize it. The literatures section is sufficiently discussed.

3. Section 3, mention the properties of the materials used “cement, sand, gravel, rubber, flyash etc”

4. Most sections of the article describe the testing procedure and the supporting discussion for each test is lacking and should be discussed in detail.

5. Results and discussions are shallow with lacking information, without any citations to justify the trends of results. Some earlier findings are focused and discussed rather than the current investigation. Discuss each parameter one by one.

6. The sentences “These experiments are under investigation now and will be submitted for publication when they are finished” are inappropriate to the conclusions.

7. Conclusions should be improved based on the new discussions.

Author Response

First of all, we thank wholeheartedly the reviewer’s putting time and effort in reviewing this article and providing comments and critics. Below are the replies.

-------------------------------------------------------------------------------------------------------------------

This article investigates the durability of decade-old crumb rubber concrete cored specimens. The important discussion sections are not clearly presented and they should improve extensively. This article requires more improvement to enhance its quality. The comments are listed below.

  1. Abstract: discuss one or two lines about the current issues of this research, the solution and what is the need for this research. The description of the investigation is shallow and should be elaborated. I suggest putting numbers within brackets (1), (2) and (3)….“The three experimental tests are: 1 the flexural stress-strain test on semi-circular disk specimens; 2 the accelerated steel-rebar corrosion test and 3 the carbonation test. In addition, the in-situ carbonation test was also carried out on the CRC bridge deck.”

Reply: Consider Abstract can have only 200 words, The content of “the current issues of this research, the solution and what is the need for this research.” is presented in the Introduction. Yes, (1), (2) and (3) are taken and revised accordingly.

  1. Introduction: I suggest adding a few lines about the issues due to crumb rubber and how it affects the environment and current methods to minimize it. The literatures section is sufficiently discussed.

Reply: The essence of this article is a material engineering research. As such, it is briefly touched on its relevance with the environment theme as stated in the beginning of the Introduction:” Crumb rubber concrete (CRC) is a composite material by adding rubber crumbs into Portland cement concrete and rubber crumbs are made by shredding old automobile tires. In this sense, CRC is a “green” material.” The further expansion on this theme is limited by the article page length as well as being seen as beyond its essence.

  1. Section 3, mention the properties of the materials used “cement, sand, gravel, rubber, flyash etc”

Reply: This study continues the article in Ref.[13] by the current authors. What mentioned here can be found in Ref.[13], which is omitted here for avoiding redundance as well as copyright issue.

  1. Most sections of the article describe the testing procedure and the supporting discussion for each test is lacking and should be discussed in detail.

Reply:Semi-circular disk bending test and Accelerated steel rebar corrosion test are unconventional and no specification or standard being established officially. Since the essence of this article is to present the findings and due to the limit on article page length, Ref.[14] to Ref.[22] are given in this article to provide limited reference on the methodology of the two experiments. Anyone interested in the two experiments can follow these references. On the other hand, the carbonation test has a standardized procedure and can be found in: <China Standard for test methods of long-term performance and durability of ordinary concrete, (GB/T 50082-2009)>, and it is added in the revision.

  1. Results and discussions are shallow with lacking information, without any citations to justify the trends of results. Some earlier findings are focused and discussed rather than the current investigation. Discuss each parameter one by one.

Reply: The first and primary task of this article is to disseminate its findings to the concrete community, and it will take as much page space as needed to accomplish this task. The analysis and discussion on those findings is the second task in this article, which may involve much broad based contents and issues, just like what the proverb say: “Rome was not built in one day”. At the same time, the effort is made to address this second task as much adequately as possible with the limit or cap on page space.

In concrete airfield pavement design, flexural elastic modulus is an important parameter. Following the reviewer’s comment here, the following paragraph is added in the Conclusion:

“Lowing flexural elastic modulus may have a profound influence on airfield pavement design. In current design cases, the flexural elastic modulus is set at a default value of 36GPa. The study conducted by Li et al. [35] indicates that, because of its low value of flexural elastic modulus at 20-30GPa range, CRC could reduce the thickness of concrete airfield pavement by 9% and increase the fatigue pavement life in a noticeable way. Table 3 in this article shows the values decrease to 10-20GPa range at a decade old time frame, it appears that CRC may be plausible for providing more benefit to concrete airfield pavement design.”

  1. The sentences “These experiments are under investigation now and will be submitted for publication when they are finished” are inappropriate to the conclusions.

Reply: Well taken, the sentence is removed 

  1. Conclusions should be improved based on the new discussions.

Reply: please see the revision.

Reviewer 2 Report

This Manuscript investigates the Mechanical and durability study of Decade Old Crumb Rubber Concrete Cored Specimens. The introduction provide sufficient background of literatures. The research design is found appropriate. The experimental test portion are covered with all the test performed in this study in details. The results are presented clearely with figures, Tables and charts. The conclusion are supported by the results obtained in this study. Are all the cited references are relevant to this research. However the following minor corrections are to be addressed before the acceptance of the manuscript.

1.Title:

The title can be Revised as follows. “Study on Mechanical and Durability Properties of a Decade Old Crumb Rubber Concrete Cored Specimens”

2. Introduction: Page 1.

Correct the citation  (Li and Li [9]) into [9]

3. Table 1 (kg/m3) – write the cube in superscript.

4. Figure 3. Start the figure captions with Upper case

5. Figure 4.  Change the figure caption as “SCDB test setup

6. Figure 5 Change the figure caption as “accelerated steel rebar corrosion test setup”

7. In CO2., bring 2 in suffix throughout the manuscript.

8. Section 4.1.1. Test program should be presented under the section of 3. Experimental Tests.

9.Figure 9. Change the figure caption as “linear and non-linear responses of SCDB test “

10. Section 4.2.1. Test program should be presented under the section of 3. Experimental Tests.

11. Section 4.3.1. Test program should be presented under the section of 3. Experimental Tests.

Author Response

First of all, we thanks wholeheartedly the reviewer’s putting time and effort in reviewing this article and providing comments and critics. Below are the replies.

---------------------------------------------------------------------------------------------------------------------

1.Title:

The title can be Revised as follows. “Study on Mechanical and Durability Properties of a Decade Old Crumb Rubber Concrete Cored Specimens”

Reply:Well received. In talking with the journal editor about the change of title.

  1. Introduction: Page 1.

Correct the citation  (Li and Li [9]) into [9]

Reply: Yes, revised accordingly.

  1. Table 1 (kg/m3) – write the cube in superscript.

Reply: Yes!revised accordingly.

  1. Figure 3. Start the figure captions with Upper case

Reply:Yes, revised accordingly.

  1. Figure 4.  Change the figure caption as “SCDB test setup“

Reply: Yes, revised accordingly.

  1. Figure 5 Change the figure caption as “accelerated steel rebar corrosion test setup”

Reply: Yes, revised accordingly.

  1. In CO2., bring 2 in suffix throughout the manuscript.

Reply: Yes, revised accordingly.

  1. Section 4.1.1. Test program should be presented under the section of 3. Experimental Tests.

Reply: Section 3 is more about of introducing the methodology of the three experiments. Test program is more about test procedure, which could be put under both Section 3 and Section 4. Consider the paragraph of Test program is short and has a direct correlation with the test results, it may be more suitable keeping it in Section 4.   

9.Figure 9. Change the figure caption as “linear and non-linear responses of SCDB test “

Reply: Yes, revised accordingly.

  1. Section 4.2.1. Test program should be presented under the section of 3. Experimental Tests.

Reply: Section 3 is more about of introducing the methodology of the three experiments. Test program is more about test procedure, which could be put under both Section 3 and Section 4. Consider the paragraph of Test program is short and has a direct correlation with the test results, it may be more suitable keeping it in Section 4.

  1. Section 4.3.1. Test program should be presented under the section of 3. Experimental Tests.

Reply: Section 3 is more about of introducing the methodology of the three experiments. Test program is more about test procedure, which could be put under both Section 3 and Section 4. Consider the paragraph of Test program is short and has a direct correlation with the test results, it may be more suitable keeping it in Section 4.

Reviewer 3 Report

The article is interesting and addresses a topic that deserves research, the durability of the material after a long-time service.

Although section 2 is very short and I am not very sure that it is worth including it as an independent section, I can understand the relevance of the paragraph and the reason why the authors made the decision of writing it as a different section.

The application of the SCDB test is very frequent in asphalt concrete, but it can also be used in Portland cement concrete as in this study. This is a simple way of indirectly subjecting the material to a tensile stress but I would like to ask some questions about it: Isn’t there a notch in the middle of the flat face of the specimen to induce the cracking? What is the test temperature? Is there any specification (minimum values) for the energies and moduli, at a specific temperature?

Regarding corrosion and carbonation tests, it is very interesting to compare the results with similar structures without crumb rubber, as they can confirm the good behavior of the solution proposed. Similarly, the analysis of the prediction models for carbonation seems a very useful contribution for developing new adjustments coefficients and improving the models. No comments about these tasks.

It would be an alternative to try to publish this article as the first part of the study and then, to go on with a second part showing the results of freeze-thaw tests and porosimetry.

A minor detail is that the paragraph after figure 8 shows some symbols that seem to be instead of Sigma and Epsilon. Please correct them.

Author Response

First of all, we thank wholeheartedly the reviewer’s putting time and effort in reviewing this article and providing comments and critics. Below are the replies.

------------------------------------------------------------------------------------------------------------------

The article is interesting and addresses a topic that deserves research, the durability of the material after a long-time service.

Although section 2 is very short and I am not very sure that it is worth including it as an independent section, I can understand the relevance of the paragraph and the reason why the authors made the decision of writing it as a different section.

Reply: Appreciate for the understanding!

The application of the SCDB test is very frequent in asphalt concrete, but it can also be used in Portland cement concrete as in this study. This is a simple way of indirectly subjecting the material to a tensile stress but I would like to ask some questions about it: Isn’t there a notch in the middle of the flat face of the specimen to induce the cracking? What is the test temperature? Is there any specification (minimum values) for the energies and moduli, at a specific temperature?

Reply: The specimens in hot mix asphalt (HMA) SCDB test does has a notch in the middle of the flat face of the specimen, which is a fracture toughness oriented experiment and analysis. For the three-point bending test given in this article, as shown in the figure below, the upper-left and upper-right portion of a beam specimen(grey regions) contributes limited load-carrying capacity. So the failure feature of a concrete beam bending test is kept by SCDB test, that the rapture occurs at the mid-span on the bottom face. The difference between HMA and concrete SCDB is that the loading speed rate for concrete SCDB test in this article is 0.03mm/second, or 1.8 mm per minute. For HMA SCDB test, the loading speed is much higher for the reason that the modulus or rigidity for HMA is much lower than concrete so the load frame can afford higher loading speed. Not like HMA, concrete is less sensitive to temperature, whereby, the issue of temperature is not given much attention at the moment.

Regarding corrosion and carbonation tests, it is very interesting to compare the results with similar structures without crumb rubber, as they can confirm the good behavior of the solution proposed. Similarly, the analysis of the prediction models for carbonation seems a very useful contribution for developing new adjustments coefficients and improving the models. No comments about these tasks.

Reply: Thanks!

It would be an alternative to try to publish this article as the first part of the study and then, to go on with a second part showing the results of freeze-thaw tests and porosimetry.

Reply: Well received! The results or numbers obtained from the freeze-thaw tests of cored CRC specimens are small, only a few lines. So, there is a need to have some analytical models going along with the experimental results so, paper lengthwise, it can make a full section.  There are too many existing concrete freeze-thaw models and we are learning them, but we feel kind of being lost and do not know which one to go with. The same for the porosity study, which is even messier.

A minor detail is that the paragraph after figure 8 shows some symbols that seem to be instead of Sigma and Epsilon. Please correct them.

Reply: Yes, revised accordingly.

Reviewer 4 Report

The authors present the study and the results of three experiments on long-aging crumb rubber concrete (CRC) samples taken from a bridge located in China, which has been in use for ten years. The research concerns an interesting subject and although they include a very small number of sample elements, which does not allow for statistical processing of the results, they can be useful for people dealing with similar topics.

1. Page 2. Abbreviation "SPA" is not previously explained in the text.

2. Page 4. Final paragraph. When describing the preparation of samples in the form of semi-circular disks for stress-strain tests, authors did not provide whether the surfaces were somehow additionally prepared before sticking the strain gauges, e.g. eliminating visible holes in the surface of the element with an adhesive (e.g. epoxy resin). Moreover, according to what dependence the lengths of the measurement bases of these strain gauges were assumed? This is of major importance for the reliability of the deformation measurement. Typically, the length of the gauge base should not be less than 3.5 times the diameter of the largest aggregate fraction. Fig. 3 clearly shows that the crumb rubbers and/or aggregate are large in relation to the length of the strain gauges.

3. Page 5. There are no dimensions at all in the test schematic shown in Fig. 4. This is unacceptable in research papers.

4. Page 5. Fig.5 and description of this drawing. There is no information as to why the NaCl concentration was assumed to be 3.5%. In addition, the usual molar concentration is used, not the percentage. According to what standards or regulations is this method of testing corrosion resistance adopted?

5. Page 6. First paragraph. Was force or deformation control applied? From the charts shown in Fig.8, it can be seen that the research was controlled by force, because there are no falling branches in these charts. The authors' commentary would be needed in this regard.

6. Page 7. Final paragraph. Typing errors occurring when transcribing the article to a pdf file. The same occurs on the next page under formula (3).

7. Pages 12-13. Conclusions. This subsection should be slightly reedited. First, it must be noted that the tests concerned only one material from one application (the bridge plate) and were made on a very small number of samples. Such were the realities, but this implies certain limitations on the possibility of a broader generalization. This means that the results are qualitatively correct and reliable, but cannot be quantified. Besides, the authors do not provide exact quantitative results in their conclusions, which is the correct approach. Second, the three conclusions that the authors formulate should be given in bulleted form (one below the other) and not in one long sentence.

Author Response

First of all, we thank wholeheartedly the reviewer’s putting time and effort in reviewing this article and providing comments and critics. Below are the replies.

---------------------------------------------------------------------

The authors present the study and the results of three experiments on long-aging crumb rubber concrete (CRC) samples taken from a bridge located in China, which has been in use for ten years. The research concerns an interesting subject and although they include a very small number of sample elements, which does not allow for statistical processing of the results, they can be useful for people dealing with similar topics.

  1. Page 2. Abbreviation "SPA" is not previously explained in the text.

Reply: Well received and revied accordingly. Superabsorbent polymer (SAP) is a water absorbing polymer.

  1. Page 4. Final paragraph. When describing the preparation of samples in the form of semi-circular disks for stress-strain tests, authors did not provide whether the surfaces were somehow additionally prepared before sticking the strain gauges, e.g. eliminating visible holes in the surface of the element with an adhesive (e.g. epoxy resin). Moreover, according to what dependence the lengths of the measurement bases of these strain gauges were assumed? This is of major importance for the reliability of the deformation measurement. Typically, the length of the gauge base should not be less than 3.5 times the diameter of the largest aggregate fraction. Fig. 3 clearly shows that the crumb rubbers and/or aggregate are large in relation to the length of the strain gauges.

Reply: Yes, the surface is polished and adhesive is applied to attach strain gauge. This may give rise to the recording fluctuation. All tried here is to insert three parallel strain gauges to reduce such fluctuation as much as possible. From the data obtained, they look reasonable.

  1. Page 5. There are no dimensions at all in the test schematic shown in Fig. 4. This is unacceptable in research papers.

Reply: Fig. 4 is rather a concept description of SCDB test.

  1. Page 5. Fig.5 and description of this drawing. There is no information as to why the NaCl concentration was assumed to be 3.5%. In addition, the usual molar concentration is used, not the percentage. According to what standards or regulations is this method of testing corrosion resistance adopted?

Reply: the accelerated steel-rebar corrosion test is not a standardized test, and 3.5% is what have been selected in previous studies and was followed in this study too.

  1. Page 6. First paragraph. Was force or deformation control applied? From the charts shown in Fig.8, it can be seen that the research was controlled by force, because there are no falling branches in these charts. The authors' commentary would be needed in this regard.

Reply: it is displacement control or velocity control loading diagram as stated in 4.1.1:” The load is in the speed control mode with the speed rate at 0.03mm/second” F is the reactively recorded total load. The original figure is force-strain response, and stress-strain response is obtained by computing Eq.(1).

  1. Page 7. Final paragraph. Typing errors occurring when transcribing the article to a pdf file. The same occurs on the next page under formula (3).

Reply: Yes, it may be addressed at the editing process

  1. Pages 12-13. Conclusions. This subsection should be slightly reedited. First, it must be noted that the tests concerned only one material from one application (the bridge plate) and were made on a very small number of samples. Such were the realities, but this implies certain limitations on the possibility of a broader generalization. This means that the results are qualitatively correct and reliable, but cannot be quantified. Besides, the authors do not provide exact quantitative results in their conclusions, which is the correct approach. Second, the three conclusions that the authors formulate should be given in bulleted form (one below the other) and not in one long sentence.

Reply: Well received. The sentence is added in the end of the Conclusion: “It should be mentioned that the results obtained here are on a small number of sampling, which does not possess any statistical significance.”

Round 2

Reviewer 1 Report

The article needs some improvement still. Most of the comments are not addressed clearly.

Author Response

First of all, we thank wholeheartedly the reviewer’s putting time and effort in reviewing this article and providing comments and critics. Below are the replies.

---------------------------------------------------------------------------------------------------

- The article needs some improvement still. Most of the comments are not addressed clearly.

Reply: The primary objective of this article is to present its novel findings to the concrete community. Such discernibility earmarks sequential issues and questions with all plausibility of intrinsic partiality, which serves the human mind charity preceding to the science development.

As for as the issue of “English language and style”, from the experience of my 40 years of writing and publishing more than 100 research articles in the top-notch English language professional journals and 20 years working/teaching in USA, the “style” bewilders those who are “bewildered”.

Reviewer 4 Report

1. The information "<China Standard for test methods of long-term performance and durability of ordinary concrete, (GB / T 50082-2009)", introduced into the text on page 6, implies the need to include this standard in references and give it an appropriate number. Such an introduction to the text, without being placed in References, is unacceptable.

2. The information given in the explanations of the authors regarding the acceptance of the concentration of 3.5% NaCl that "the accelerated steel-rebar corrosion test is not a standardized test, and 3.5% is what have been selected in previous studies" should be introduced to the text of the article. Just explaining to the reviewer in the cover letter is insufficient.

Author Response

First of all, we thank wholeheartedly the reviewer’s putting time and effort in reviewing this article and providing comments and critics. Below are the replies.

---------------------------------------------------------------------

  1. The information "<China Standard for test methods of long-term performance and durability of ordinary concrete, (GB / T 50082-2009)", introduced into the text on page 6, implies the need to include this standard in references and give it an appropriate number. Such an introduction to the text, without being placed in References, is unacceptable.

Reply: From the experience of my 40 years of writing and publishing more than 100 research articles in the top-notch English language professional journals, I may claim that the prevailing scenario is that the article reference section precludes the standard(s) cited in the context of the article worldwide with the exception in China, which somehow, allures to mandate as referred. As trained professionally in the USA, the vis-à-vis compliance is to insist on the worldwide earmarking.   

  1. The information given in the explanations of the authors regarding the acceptance of the concentration of 3.5% NaCl that "the accelerated steel-rebar corrosion test is not a standardized test, and 3.5% is what have been selected in previous studies" should be introduced to the text of the article. Just explaining to the reviewer in the cover letter is insufficient.

Reply: Well received and revised according. “the accelerated steel-rebar corrosion test is not a standardized test, and 3.5% is what have been selected in previous studies" is added to the article.
